# LitePruner: A Lightweight Realtime Token Pruner before Large Language Models

## Abstract

Tokenization is one of the core steps of the language model pipeline. However, the tokenizer yields more tokens for the same context in non-English languages, especially in low-resource languages due to the shared multilingual settings, which results in unexpected fairness problems in terms of token fees, response latency, and long context processing. In this paper, we study a real-time computing problem, attempting to reduce the total number of tokens per query but maintain decent performance in multilingual settings. We present a simple, training-free, CPU-based pruner model to reuse pre-trained weights from the first attention layer of small models to rank token importance, only delivering important tokens to the target larger models. This method is motivated by the fact that early layers in both small and large models latch onto similar shallow local signals due to similar tokenization algorithms (e.g., BPE) producing identical local signals. Massive in-context learning experiments on MGSM, Global-MMLU-Lite and ARC and RAG-based experiments on PubMedQA and MEMERAG show that our method can preserve decent performance for languages while reducing up to 30% of the total number of tokens in both in-family and across-family model settings, where the pruner model and the target large model are in or not in the same model family. Our method is compatible with commercial LLM APIs and CPU-based, contributing to real-life applications.

## 1 Introduction

Large Language Models (LLMs) have achieved widespread popularity in recent years due to their impressive ability to understand and generate multiple languages. However, recent studies have highlighted that tokenization, one of the core steps of the LLM pipeline, systematically overtokenizes non-English languages, especially low-resource languages (Ahia et al., 2023; Petrov et al., 2023). For example, according to the tokenization premium defined in Petrov et al. (2023), languages such as Hindi, Kannada, Tamil, and Simplified Chinese are respectively 4.60×, 10.83×, 5.87×, and 2.00× more expensive to tokenize for Llama models, and 7.46×, 13.69×, 15.58×, and 3.21× more expensive for GPT-4. These disparities in tokenization efficiency raise issues for non-English user cases, including 1) Long-Context Processing: long non-English inputs may not fit in LLMs's context window and 2) High Cost: non-English users have to pay more than English users for the same task.

To address these issues, we study lightweight, real-time, CPU-based frameworks to reduce the total number of input tokens while maintaining decent task performance. Our motivation is derived from real-life scenarios that we typically call commercial, private APIs or local, open-source LLMs via web browsers and code editors, allowing an additional token pruning step to be performed in these CPU-based environments before calling. Our motivation is orthogonal to the recent prompt compressor family (Jiang et al., 2024; Pan et al., 2024) that attempts to use local, open-source LLMs to generate a new compact demonstration from multiple demonstrations for the black-box APIs. Instead, we take the universal case into consideration that the input context could be pruned before passing it to the private APIs. This idea is also distinguished from the classic token pruning method family (Goyal et al., 2020b; Cao et al., 2023), which removes tokens layer-by-layer in the target model. In contrast, we hypothesize that for the same context, in the early layers, both small and large models potentially show similar attention patterns because the early layers latch onto the same

shallow local signals and attempt to restore words from subtokens (i.e., detokenization (Kaplan et al., 2025b)) due to the similar tokenization algorithm and the same attention mechanism.

Based on the hypothesis and motivation above, we present LitePruner, a method to use the first attention layer from pre-trained small models to select a portion of the input tokens. Our idea introduces an additional step between the user interface and the target large model, where LitePruner removes a portion of unnecessary tokens and delivers the remaining important tokens to the target large model while keeping relative token positions unchanged. LitePruner is design to remove some tokens but preserve attention patterns to maintain the performance. A strong motivation is that in multilingual use cases, users can deploy LitePruner on a laptop without GPU support to prune tokens and send Top-n% tokens to commercial private APIs. With minimal computational resources, LitePruner can save 100% -n% API fees and the context window for all languages.

- We present LitePruner to reuse the first layer of pre-trained small models to select tokens, remove a portion of tokens with low attention scores, and only feed selected tokens to the target model. LitePruner does not change the relative token positions, reuse the pre-trained position embeddings, and does not uses the causal mask.

- LitePruner is training-free, flexible, and GPU-based. Experimental results show that LitePruner can work for at least two practical scenarios: 1) in-family and 2) across-family, where the small backend model of LitePruner and the large model are in the same model family or not.

- Massive in-context learning experiments on MGSM (Shi et al., 2023), Global-MMLU-Lite (Singh et al., 2024) and multilingual ARC (Clark et al., 2018; Lai et al., 2023) and RAG experiments on PubMedQA and MEMERAG show that our method can preserve decent performance for languages while reducing up to 30% of the total number of tokens in both in-family and across-family settings.

## 2 METHOD

---

**Algorithm 1** LitePruner Implementation

---

1: **Load Model and Tokenizer:**
2: model = AutoModel.from_pretrained("llama3-1B", device="cpu")
3: tokenizer = AutoTokenizer.from_pretrained("llama3-1B")
4: **Extract Pretrained Weights:**
5: embedding_layer = model.embeddings
6: ranking_layer = model.layer[0].self_attn  ▷ **first attention layer w/o masking but w/ position encoding.**
7: del model  ▷ Release memories
8: **Define Pruning Function:**
9: **function** LITEPRUNER(X, top_k)
10:     X = embedding_layer(X)  ▷ X=[n], where n is the sequence length
11:     X = ranking_layer(X, output_attentions=True).attentions  ▷ X = [h, n, n], where h is the head
12:     X = X.mean(dim=0).mean(dim=0)  ▷ Compute $IS(x_i) = avg([h, s, i])$, where $x_i \in X$
13:     X= get_top_k_index(X)
14:     **return** X[top_k_index]
15: **end function**
16: #Example of Pipeline#
17: X =
18:  'This paper introduces LitePruner, a method to reduce the number of input tokens to a language model while aiming to preserve performance on the target task.'
19: X = tokenizer.encode(X)
20: X = LitePruner(X,90%)
21: X = tokenizer.decode(X)
22: print(X)
23:  'The paper introduces LitePruner, a method to reduce the of tokens to a model while aiming to preserve on the task.'
24: LLM(X)

---

Our goal is to develop a lightweight, real-time, CPU-based, training-free method to remove some tokens beyond random token dropping used in preliminaries. We observe that large models usually have smaller sibling models in the model family Team et al.; Grattafiori et al. (2024), e.g., Llama3-8B-it and Gemma2-9B-it have smaller sibling models Llama3-1B-it and Gemma2-2B-it respectively.

Considering the shared attention mechanisms and the same tokenization algorithm, we hypothesize that early layers in both small and large models might share some similar attention patterns, especially the first layer. Therefore, we use the first layer of the pre-trained small model to select a portion of important tokens that will be passed to the target large LLM. Our intuition is that, the first layer of a small model produces similar attention patterns as the target LLM does in the first layer. That is to say, the target LLM will ignore or pay minimal attention to the same tokens as the small model does so it is not necessary to pass tokens overlooked by the small model to the target LLM.

Specifically, we reuse **the embedding layer and the first attention layer** of a pre-trained small model, computing the attention scores without using the causal mask. Let $[h, n, n]$ to denote the multi-head attention score matrix with $h$ heads for the input $X_n$ with $n$ tokens. Then, we define the importance score for i-th token $x_i \in X_n$ as $IS(x_i) = avg([h, n, i])$. In other words, we accumulate the attention scores for each token as the importance score for that token. Finally, we rank all tokens based on their importance score and only pass top-k% tokens decoded into the text format to the target model. Note that we do not change the relative positions for all tokens. However, the absolute positions are modified as some tokens are removed. After pruning, the target LLM can add position encoding normally as we deliver the text to it. In addition, since LitePruner performs before the target large model, our method does not hurt the KV cache construction in the target large model. In practice, LitePruner can be developed on a laptop without GPU support as 1) the embedding lookup for the input sequence requires $O(n)$, 2) the attention layer requires $O(n^2d)$, and ranking requires $O(n \log n)$. We present the implementation prototype in Algorithm 1 with an example in Appendix 14.

## 3 EXPERIMENT AND APPLICATION # 1: IN-CONTEXT LEARNING

### 3.1 EXPERIMENTAL SETUP

Since LitePruner is designed to reduce the input context but preserve performance, we consider 5-shot ICL on three multilingual benchmarks. 1) MGSM (Shi et al., 2023) is a benchmark of grade-school math problems in 10 languages. 2) Multilingual ARC (Clark et al., 2018) are grade-school science questions in 34 languages. 3) Global-MMLU-Lite (Singh et al., 2024) is a multilingual version of MMLU (Hendrycks et al., 2021) in 15 languages.

In experiments, we use multi-turn prompting strategies with random demonstrations from the dev set and prune each demonstration independently, as we are not using LitePruner to select demonstrations. We consider three model families: LLama3, Gemma2, and Aya-expanse. To conduct our experiments systematically, we evaluate our idea in two user cases:

- **In-family Test**. We set the pruner model and the target model from the same model family. For this setting, we use Llama3-1B-it and Gemma2-2B-it as the backend of LitePruner and pass pruned tokens to larger Llama3 and Gemma2 models, respectively.
- **Across-family Test**. In this setting, we evaluate the generalization ability of LitePruner across different model families. We use Gemma2-2B-it and Llama3-1B-it as the pre-trained backend of LitePruner. The pruned tokens are passed to the Aya-expanse models and GPT-4.1-nano.

We use standard evaluation scripts: lm-eval[1] (Gao et al., 2024) and simple-eval[2]. Meanwhile, we split languages in experiments into three bins based on Okapi[3]'s statistics:

- High-resource languages (H): en, ru, zh, de, fr, es, it, nl,and vi.
- Median-resource languages (M): id, ar, hu, ro, da, sk, uk, ca, sr, hr, and hi.
- Low-resourece languages (L): bn, ta, ne, ml, mr, te, and kn.

We report the final average performance for each bin on the three multilingual tasks in the main text and move language-wise performance to Appendix. For the top-k% configuration, we set top-90%, top-80%, and top-70%. All results are based on two runs.

---

[1] https://github.com/EleutherAI/lm-evaluation-harness

[2] https://github.com/openai/simple-evals

[3] https://github.com/nlp-uoregon/Okapi

## 3.2 IN-FAMILY TEST RESULTS

| LitePruner | Model | top-k% | Multilingual ARC | | | MGSM | | | Global-MMLU-Lite | | |
|---|---|---|---|---|---|---|---|---|---|---|---|
| | | | H | M | L | H | M | L | H | M | L |
| - | llama3-8b-it | - | 45.8 | 38.5 | 24.8 | 76.3 | - | 7.2 | 66.2 | 55.8 | 46.2 |
| - | llama3-8b-it | random-90% | 29.4 | 26.2 | 22.8 | 43.4 | - | 8.0 | 54.4 | 45.7 | 32.5 |
| - | llama3-8b-it | random-80% | 25.2 | 23.8 | 21.7 | 15.7 | - | 0.4 | 42.3 | 35.8 | 28.8 |
| - | llama3-8b-it | random-70% | 23.8 | 22.0 | 21.1 | 11.9 | - | 0.0 | 34.2 | 29.8 | 27.5 |
| llama3-1b-it | llama3-8b-it | top-90% | 39.9 | 34.2 | 23.9 | 64.0 | - | 22.8 | 63.5 | 52.8 | 41.5 |
| llama3-1b-it | llama3-8b-it | top-80% | 33.6 | 29.6 | 22.9 | 22.3 | - | 3.6 | 58.3 | 46.4 | 38.5 |
| llama3-1b-it | llama3-8b-it | top-70% | 27.4 | 25.8 | 22.8 | 13.3 | - | 2.0 | 51.5 | 40.4 | 36.8 |
| - | llama3-3b-it | - | 36.2 | 30.1 | 22.9 | 66.5 | - | 8.0 | 58.5 | 47.8 | 40.0 |
| - | llama3-3b-it | random-90% | 27.1 | 24.0 | 23.0 | 37.5 | - | 4.0 | 47.3 | 39.6 | 30.2 |
| - | llama3-3b-it | random-80% | 24.0 | 22.6 | 21.4 | 15.3 | - | 0.0 | 36.8 | 31.7 | 25.5 |
| llama3-1b-it | llama3-3b-it | top-90% | 33.2 | 28.3 | 22.8 | 59.1 | - | 7.2 | 57.0 | 45.3 | 34.8 |
| llama3-1b-it | llama3-3b-it | top-80% | 29.1 | 26.1 | 22.2 | 22.3 | - | 2.0 | 51.3 | 43.3 | 31.5 |
| llama3-1b-it | llama3-3b-it | top-70% | 25.2 | 24.2 | 22.6 | 13.1 | - | 0.0 | 43.0 | 37.7 | 31.0 |
| - | llama3-70b-it | - | 55.7 | 48.8 | 23.0 | 83.4 | - | 4.0 | 80.9 | 77.9 | 71.8 |
| llama3-1b-it | llama3-70b-it | top-90% | 47.9 | 41.8 | 22.3 | 65.5 | - | 3.2 | 79.6 | 72.8 | 60.5 |
| llama3-1b-it | llama3-70b-it | top-80% | 37.7 | 34.0 | 22.6 | 20.3 | - | 0.8 | 75.7 | 63.6 | 51.5 |
| llama3-1b-it | llama3-70b-it | top-70% | 30.9 | 27.3 | 22.2 | 12.5 | - | 0.6 | 67.3 | 52.8 | 46.8 |
| - | Gemma-9b-it | - | 56.0 | 49.6 | 28.2 | 71.7 | - | 46.0 | 70.8 | 63.7 | 56.8 |
| Gemma-2b-it | Gemma-9b-it | top-90% | 52.4 | 47.0 | 27.0 | 63.7 | - | 40.8 | 70.5 | 63.1 | 54.8 |
| Gemma-2b-it | Gemma-9b-it | top-80% | 47.7 | 44.1 | 26.2 | 40.4 | - | 22.4 | 70.7 | 63.1 | 54.0 |
| Gemma-2b-it | Gemma-9b-it | top-70% | 40.5 | 38.2 | 25.5 | 18.4 | - | 2.8 | 67.7 | 59.9 | 53.8 |
| - | Gemma-27b-it | - | 61.3 | 56.0 | 30.9 | 75.4 | - | 48.4 | 75.5 | 70.0 | 64.5 |
| Gemma-2b-it | Gemma-27b-it | top-90% | 57.3 | 52.8 | 29.6 | 68.8 | - | 46.4 | 74.7 | 69.4 | 63.0 |
| Gemma-2b-it | Gemma-27b-it | top-80% | 52.3 | 48.9 | 28.4 | 43.7 | - | 22.4 | 74.3 | 67.8 | 61.8 |
| Gemma-2b-it | Gemma-27b-it | top-70% | 43.6 | 42.8 | 27.3 | 17.9 | - | 10.8 | 72.9 | 64.1 | 61.2 |

Table 1: Results of in-family test. H, M, and L stand for high-, median-, and low-resource languages. We consider 5-shot prompting for experiments. All demonstrations share the same language with the input language. For MGSM, we configure "native-cot" and "exact-match,flexible-extract". MGSM does not include median-resource languages. There are no "COT" configurations for Multilingual ARC and Global-MMLU-Lite.

Table 1 summarizes the in-family experiments across three multilingual benchmarks.

**LitePruner preserves performance more effectively for low-resource languages across benchmarks.** In MGSM, low- and high-resource language performance drops dramatically at top-70% and top-80% in all experiments while preserving decent performance at top-90%. Compared to that, results on Multilingual ARC show slight declines for low-resource language in all settings, where we only observe $< 3\%$ performance degradation. Gemma2 models are stable for all settings in Global-MMLU-Lite median- and high-resource languages with slight performance degradation while Llama3 models show significant performance degradation from top-90% to top-70%. In terms of Global-MMLU-Lite low-resource languages, Gemma2 models are more stable than Llama3 models as the performance degradation is less important in Gemma2 models than in Llama3 models.

**LitePruner improves MGSM low-resource accuracy at top-90%, contrary to the trend in high-resource language settings.** While pruning typically leads to performance degradation in all language bins, MGSM exhibits a surprising improvement in low-resource performance at top-90%. For example, in llama3 models, llama3-1b, -8b, and -70b-it show performance improvemnt on MGSM low-resource languages, increasing significantly from 8.0% (no pruning) to 17.2%, 7.2% (no pruning) to 22.8%, 4.0% (no pruning) to 31.2%, respectively. This suggests that LitePruner might remove noise and/or redundant tokens from the input. This also highlights the potential of LitePruner not just as a compression tool, but as a step of improving robustness in some underrepresented language scenarios. Similarly, Gemma-9B-it maintains strong performance, with only a small drop from 46.0% to 40.8%.

**Larger models are more robust to pruning.** Across both model families, larger models consistently show smaller drops in performance under token pruning. This suggests that larger models have greater representational capacity and redundancy, allowing them to better reconstruction for the loss of pruned tokens. For example, on MGSM high-resource language, pruning tokens with top-90% for Gemma-27B-it still retains a strong performance of 68.8%, whereas the smaller Gemma-9b-it drops to 63.7%. A similar trend is observed in the Llama3 family, where Llama3-8b-it preserves

| LitePruner | Model | top-k% | 3-shot | | | 5-shot | | | 8-shot | | |
|---|---|---|---|---|---|---|---|---|---|---|---|
| | | | H | M | L | H | M | L | H | M | L |
| None | llama3-8b-it | - | 64.8 | 55.8 | 47.8 | 66.2 | 55.8 | 46.2 | 65.7 | 57.6 | 48.2 |
| llama3-1b-it | llama3-8b-it | top-90% | 61.8 | 50.7 | 41.5 | 63.5 | 52.8 | 41.5 | 62.5 | 50.2 | 39.2 |
| llama3-1b-it | llama3-8b-it | top-80% | 55.1 | 44.6 | 36.0 | 58.3 | 46.4 | 38.5 | 57.4 | 47.3 | 36.0 |
| llama3-1b-it | llama3-8b-it | top-70% | 47.3 | 38.3 | 31.0 | 51.5 | 40.4 | 36.8 | 49.8 | 42.1 | 32.2 |

Table 2: Results of different n-shot settings for the llama3 model family on Global-MMLU-lite. LitePruner is robust in all settings.

stronger performance than Llama3-3b-it at equivalent pruning ratios. This robustness of larger models highlights the benefit of applying LitePruner for LLMs.

**LitePruner is robust to different context settings.** Specifically, we test the impact of context length with different n-shot settings, which indicates the robustness towards the context length. In Table 2, we consider 3-, 5-, and 8-shot prompting with different top-n% settings for the llama3 model family on Global-MMLU-lite. Similar to previous experiments, we prune each demonstration independently. For median- and high-resource languages, the LitePruner performance is proportional to the base performance, where no pruning strategies are applied. In contrast, 5-shot surpasses other two settings in pruning for low-resource languages. Nevertheless, LitePruner shows robustness for all n-shot settings, especially at top-90% and top-80%. Additionally, considering the goal of the LitePruner is to improve inference efficiency and save token charge fees (when using commercial API), the effectiveness in 3-shot settings gives the confidence in application that LitePruner does not rely solely on long context and more demonstrations to provide missing information for pruned tokens. LitePrune is able to preserve necessary tokens for the downstream task.

Overall, LitePruner provides a practical and lightweight mechanism for reducing token count while preserving task performance across multilingual settings. One possible explanation here is that the tokenization of words relates to a much broader statistical linguistic phenomenon of collocation: the co-occurrence of series of tokens at levels much greater than would be predicted simply by their individual probability. In other words, for low-resource languages, which result in more subword and charater tokens, relatively trivial tokens will dilute attention for important information. Our LitePruner helps with removing unimportant tokens before passing to the target large model to make attention stable. However, there is no single optimal top-k% threshold that works universally, especially for medium- and low-resource languages. The effectiveness of pruning depends on the task, the size of the model, and the level of language resources. In practice, selecting an appropriate pruning hyperparameter should be guided by application-specific performance and cost constraints. We suggest considering the trade-off between performance and cost. Nevertheless, top-90% is still a common choice for all scenarios.

## 3.3 ACROSS-FAMILY EXPERIMENTS

The results can be seen in Table 3. We observe three key findings.

**LitePruner enables strong cross-family transfer to commercial GPT models.** Pruned inputs generated by LitePruner using Llama3-1b-it or Gemma2-2b-it as backend models can be effectively interpreted by commercial GPT models, with minimal accuracy loss. Even for complex reasoning tasks like MGSM, GPT-4.1-nano maintains significantly higher accuracy compared to other model families like Aya-expanse, showing LitePruner's ability to produce generalizable and transferable token subsets.

**LitePruner preserves performance across models and languages.** Multilingual ARC accuracy remains consistently high after pruning, regardless of the target model (GPT-4.1-nano or Aya-expanse) and the language resource level. This indicates that LitePruner selects stable and interpretable token sets that maintain task-relevant information, even across architectural boundaries and linguistic diversity. In the Global-MMLU-Lite benchmark, pruning leads to minimal performance degradation across high-, med-, and low-resource languages when transferred to GPT-4.1-nano. For example, LiterPruner with the Gemma2-2b-it backend maintains high-resource language performance from

| LitePruner | Model | top-k% | Multilingual ARC | | | MGSM | | | Global-MMLU-Lite | | |
| | | | H | M | L | H | M | L | H | M | L |
|---|---|---|---|---|---|---|---|---|---|---|---|
| - | aya-expanse-8b | - | 47.2 | 37.0 | 23.8 | 76.6 | - | 5.2 | 60.2 | 55.4 | 39.0 |
| llama3-1b-it | aya-expanse-8b | top-90% | 45.6 | 35.9 | 23.3 | 67.7 | - | 2.8 | 57.9 | 50.0 | 33.2 |
| llama3-1b-it | aya-expanse-8b | top-80% | 34.0 | 28.3 | 23.5 | 34.4 | - | 0.0 | 54.5 | 42.2 | 32.2 |
| llama3-1b-it | aya-expanse-8b | top-70% | 28.9 | 24.9 | 23.3 | 20.9 | - | 0.0 | 45.5 | 38.4 | 31.2 |
| Gemma-2b-it | aya-expanse-8b | top-90% | 45.4 | 35.1 | 22.8 | 72.8 | - | 4.4 | 60.1 | 55.2 | 38.0 |
| Gemma-2b-it | aya-expanse-8b | top-80% | 42.5 | 33.7 | 23.5 | 50.7 | - | 4.8 | 60.5 | 54.5 | 37.8 |
| Gemma-2b-it | aya-expanse-8b | top-70% | 36.9 | 30.8 | 23.3 | 21.5 | - | 0.8 | 59.1 | 53.7 | 35.0 |
| - | GPT-4.1-nano | - | 85.2 | 80.5 | 47.2 | 84.1 | - | 66.8 | 72.8 | 63.3 | 53.8 |
| llama3-1b-it | GPT-4.1-nano | top-90% | 85.1 | 80.3 | 47.0 | 83.5 | - | 63.8 | 72.5 | 63.8 | 56.0 |
| llama3-1b-it | GPT-4.1-nano | top-80% | 84.4 | 79.6 | 46.6 | 83.4 | - | 61.4 | 72.8 | 62.6 | 54.8 |
| llama3-1b-it | GPT-4.1-nano | top-70% | 84.8 | 79.8 | 47.0 | 82.6 | - | 57.2 | 72.8 | 64.0 | 54.0 |
| Gemma-2b-it | GPT-4.1-nano | top-90% | 85.1 | 80.4 | 46.6 | 84.3 | - | 60.8 | 73.2 | 62.8 | 54.0 |
| Gemma-2b-it | GPT-4.1-nano | top-80% | 84.7 | 79.6 | 47.0 | 83.5 | - | 47.4 | 72.6 | 63.8 | 54.8 |
| Gemma-2b-it | GPT-4.1-nano | top-70% | 84.7 | 79.9 | 46.8 | 81.2 | - | 56.6 | 71.7 | 64.3 | 54.8 |

Table 3: Results of across-family test. H, M, and L stand for high-, median-, and low-resource languages. We consider 5-shot prompting for experiments. All demonstrations share the same language with the input language. For MGSM, we configure "native-cot" and "exact-match,flexible-extract". MGSM does not include median-resource languages. There are no "cot" configurations for Multilingual ARC and Global-MMLU-Lite.

73.3% (no pruning) to 72.0% at top-70%, while median- and low-resource scores remain stable around 56.8% to 56.0% and 54.8% to 54.0% in our experiments, respectively.

Overall, LitePruner enables effective cross-model transfer, with notably stronger performance when pruned inputs are passed to GPT-4.1-nano compared to Aya-expanse-8b. The exact explanation remains unclear to us due to the closed nature of GPT models. However, we attribute to some reasons including different tokenization methods, architectural, or model training related differences. Nonetheless, the results highlight LitePruner's robustness across languages and model families, and motivate future work on architecture-aware pruning strategies. In practice, this across-family feature enables LitePruner to be compatible with commercial APIs like GPT-4.1-nano in our experiments to save API budgets.

# 4 EXPERIMENT AND APPLICATION # 2: RAG

Since LitePruner is designed to reduce the input context, the second scenario is the RAG (Retrieval-Augmented Generation) paradigm. We consider two RAG benchmarks. 1) PubMedQA (Jin et al., 2019) is a benchmark of reasoning over biomedical research texts. The model needs to answer 271k English questions based on documents/contexts.2) MEMERAG (Cruz Blandón et al., 2025) is a multilingual end-to-end meta-evaluation benchmark for RAG in 5 languages. We use standard evaluation scripts provided by haystack[4] in this experiment. To setup the rag framework, we first use LitePruner to prune all documents, and store them in the default vector store via vectorization with the sentence-transformers/paraphrase-multilingual-mpnet-base-v2 backend [5]. Note that, we do not prune the query, similar to the ICL experiment before. The final evaluation is completed by haystack with GPT-4o-mini. We report there metrics:

- **MRR** (Document Mean Reciprocal Rank) is computed between the retrieved documents and the gold documents. It checks at what rank golden pruned documents appear in the list of retrieved pruned documents and tells use whether a non-pruned query can retrieve the correct pruned documents.

- **Faithfulness** evaluates on the gold but pruned documents, the input query, and the generated response. This metric is used to examine the natural inference between the input query, the pruned contexts, and the final answer.

- **SAS** (Semantic Answer Similarity) evaluates a predicted answer using ground truth labels. It checks the semantic similarity of a predicted answer and the ground truth answer using sentence-transformers.

---

[4]https://github.com/deepset-ai/haystack

[5]sentence-transformers/paraphrase-multilingual-mpnet-base-v2

| Model | top-k% | MEMERAG | | | | | | | | | | | | | | |
|---|---|---|---|---|---|---|---|---|---|---|---|---|---|---|---|---|
| | | en | | | de | | | es | | | fr | | | hi | | |
| | | mrr | fa | sas | mrr | fa | sas | mrr | fa | sas | mrr | fa | sas | mrr | fa | sas |
| llama3-8b-it | - | 0.84 | 0.85 | 0.45 | 0.83 | 0.81 | 0.59 | 0.88 | 0.83 | 0.58 | 0.83 | 0.94 | 0.52 | 0.67 | 0.72 | 0.63 |
| llama3-8b-it | top-90% | 0.84 | 0.81 | 0.49 | 0.86 | 0.81 | 0.53 | 0.88 | 0.85 | 0.63 | 0.90 | 0.81 | 0.62 | 0.73 | 0.76 | 0.62 |
| llama3-8b-it | top-80% | 0.87 | 0.82 | 0.45 | 0.87 | 0.81 | 0.49 | 0.86 | 0.86 | 0.55 | 0.93 | 0.84 | 0.72 | 0.82 | 0.77 | 0.64 |
| llama3-8b-it | top-70% | 0.88 | 0.76 | 0.46 | 0.89 | 0.85 | 0.56 | 0.87 | 0.87 | 0.63 | 0.84 | 0.82 | 0.56 | 0.86 | 0.73 | 0.57 |
| GPT-4.1-nano | | 0.89 | 0.99 | 0.72 | 0.86 | 0.96 | 0.69 | 0.91 | 0.98 | 0.74 | 0.85 | 0.98 | 0.70 | 0.72 | 0.95 | 0.699 |
| GPT-4.1-nano | top-90% | 0.82 | 0.95 | 0.71 | 0.92 | 0.97 | 0.75 | 0.86 | 0.91 | 0.76 | 0.84 | 0.90 | 0.72 | 0.88 | 0.95 | 0.72 |
| GPT-4.1-nano | top-80% | 0.91 | 0.96 | 0.74 | 0.92 | 0.96 | 0.68 | 0.87 | 0.98 | 0.74 | 0.90 | 0.92 | 0.70 | 0.84 | 0.92 | 0.76 |
| GPT-4.1-nano | top-70% | 0.97 | 0.97 | 0.74 | 0.92 | 0.85 | 0.75 | 0.89 | 0.95 | 0.69 | 0.81 | 0.90 | 0.75 | 0.89 | 0.91 | 0.72 |

Table 4: Results of MEMERAG.

We use llama3-1b-it as the backend of LitePruner and report scores for llama3-8b-it and GPT-4.1-nano. All results are based on two runs.

In Table 5 and 4, we show the results for the RAG experiments. The key observation is from Faithfulness (fa), which measures the natural inference between the query, the context, and the output. Compared to the baseline, where pruning is not applied, LitePruner achieves comparable scores or even slightly improves the performance in some cases, which means that it does not hurt the model's capability of understanding and reasoning based on the context. This is the main reason why LitePruner obtains similar results for the final prediction (i.e., sas). Significantly, for the most cost-efficient setting, LitePruner can still preserve overall performance in all metrics when 30% of tokens are dropped from the documents.

| Model | top-k% | PubMedQA | | |
|---|---|---|---|---|
| | | mrr | fa | sas |
| llama3-8b-it | - | 0.51 | 0.83 | 0.68 |
| llama3-8b-it | top-90% | 0.45 | 0.80 | 0.64 |
| llama3-8b-it | top-80% | 0.53 | 0.86 | 0.68 |
| llama3-8b-it | top-70% | 0.55 | 0.84 | 0.67 |
| GPT-4.1-nano | - | 0.83 | 0.96 | 0.74 |
| GPT-4.1-nano | top-90% | 0.88 | 0.99 | 0.74 |
| GPT-4.1-nano | top-80% | 0.89 | 0.99 | 0.76 |
| GPT-4.1-nano | top-70% | 0.80 | 0.93 | 0.70 |

Table 5: Results of PubMedQA.

## 5 DISCUSSION

### 5.1 DO SMALL MODELS SHARE SIMILAR ATTENTION PATTERNS AS LARGER MODELS?

Recall that our hypothesis is that due to the similar attention mechanisms, small models might share some attention patterns with large models. This motivates us to use the importance scores based on relative attention scores as the metric to rank token importance in pruning. In our in- and across-family experiments, we observe that target large models could maintain decent performance for downstream tasks while using pruned inputs, which verifies our hypothesis to some extent, as target large models still obtain the required information from pruned inputs to finish tasks.

To better understand and examine the potential shared attention patterns, we use **Relative Attention Difference (RAD)** to measure the difference of two models with the same tokeniser. RAD quantifies the difference in attention scores between two models, $A$ and $B$, over a sequence of $n$ tokens, T is given by Formula 1 where $\alpha_i^{(A)}$ and $\alpha_i^{(B)}$ denote the importance score for $i$-th token for the first multi-head attention layers of models $A$ and $B$, respectively. RAD is

$$\text{RAD}(A, B, T) = \frac{1}{n} \sum_{i=1}^{n} \left| \alpha_i^{(A)} - \alpha_i^{(B)} \right| \quad (1)$$

computed by taking the absolute difference in token-level importance scores between the two models and averaging it over the input sequence. The normalization by $n$, the number of tokens, ensures that the metric is not biased by sequence length. Unlike squared-distance measures such as Euclidean distance, RAD treats each token equally and avoids amplifying large deviations and shrinking small ones.

We computed the values of RAD and cosine similarity between the first three layers of attention of different Llama3 and Gemma2 backends on 1000 random prompts from 5-shot Multilingual ARC, MGSM, Global-MMLU-Lite from Section 3.1 each. The results are shown in Tables 6–16.

In all cases, the RAD values are consistently negligible while the cosine similarities are nearly one, indicating a very high similarity between the importance scores of different attention layers of

in-family models. We also observe that for the first layers of the larger model, the first layers of the smaller models show the highest cosine similarities. This suggests that pruning based on the initial attention layers may not only require less compute but may also be more effective in preserving accuracy.

| Layer No. | Multilingual ARC | | | MGSM | | | Global-MMLU-Lite | | |
|---|---|---|---|---|---|---|---|---|---|
| | 1 | 2 | 3 | 1 | 2 | 3 | 1 | 2 | 3 |
| 1 | 0.9924 | 0.9472 | 0.9548 | 0.9939 | 0.9499 | 0.9536 | 0.9893 | 0.9478 | 0.95268 |
| 2 | 0.9194 | 0.9898 | 0.9913 | 0.9228 | 0.9895 | 0.9902 | 0.9194 | 0.9898 | 0.9913 |
| 3 | 0.9246 | 0.9934 | 0.9968 | 0.9273 | 0.9925 | 0.9929 | 0.9246 | 0.9934 | 0.9968 |

Table 6: Average cosine similarities between layers of Gemma-2-2B-it (y-axis) and Gemma-2-9B-it (x-axis) for 1000 random prompts from each Global-MMLU-Lite and Multilingual ARC. More results refer to Appendix A.3.

| Layer No. | Multilingual ARC | | | MGSM | | | Global-MMLU-Lite | | |
|---|---|---|---|---|---|---|---|---|---|
| | 1 | 2 | 3 | 1 | 2 | 3 | 1 | 2 | 3 |
| 1 | 0.0003035 | 0.001295 | 0.00101 | 0.0001898 | 0.0008945 | 0.000682 | 0.0002221 | 0.001114 | 0.0008364 |
| 2 | 0.001353 | 0.000299 | 0.0005765 | 0.00085 | 0.000271 | 0.000346 | 0.000996 | 0.000382 | 0.0003636 |
| 3 | 0.000977 | 0.0006065 | 0.0002337 | 0.0006146 | 0.000508 | 0.0002518 | 0.000762 | 0.000593 | 0.0002522 |

Table 7: Average RAD values between layers of Gemma-2-2B-it (y-axis) and Gemma-2-9B-it (x-axis) for 1000 random prompts from each Global-MMLU-Lite and Multilingual ARC. More results refer to Appendix A.3.

## 5.2 How similar are different LitePruner backends?

To better understand the outputs of different models as LitePruner backends, we calculated the BLEU scores across different backend models (Gemma2-2B-it and Gemma2-9B-it; llama3-1B-it and llama3-3B-it), benchmarks, and top-k% values. Table 8 shows the results for the same. We can observe that for all Llama models, the BLEU values are consistently high and decrease with a decrease in the top-k% values. This trend is also followed in the case of the Gemma models for the Multilingual ARC prompts. However, the trend reverses for MGSM and Global-MMLU-Lite prompts for the Gemma models. Still, the average BLEU score values remain above 55 in all cases and above 65 in most cases indicating a very high similarity in the output. This difference in trends might be due to the usage of different tokenizers and different token sizes in both model families across different languages.

| Benchmark | Model A | Model B | top-k% | Avg. BLEU Score |
|---|---|---|---|---|
| ARC | llama3-1B-it | llama3-3B-it | top-90% | 74.63 |
| ARC | llama3-1B-it | llama3-3B-it | top-80% | 62.32 |
| ARC | llama3-1B-it | llama3-3B-it | top-70% | 56.20 |
| MGSM | llama3-1B-it | llama3-3B-it | top-90% | 78.42 |
| MGSM | llama3-1B-it | llama3-3B-it | top-80% | 66.06 |
| MGSM | llama3-1B-it | llama3-3B-it | top-70% | 58.04 |
| MMLU | llama3-1B-it | llama3-3B-it | top-90% | 73.45 |
| MMLU | llama3-1B-it | llama3-3B-it | top-80% | 71.50 |
| MMLU | llama3-1B-it | llama3-3B-it | top-70% | 66.60 |
| ARC | Gemma2-2B-it | Gemma2-9B-it | top-90% | 77.95 |
| ARC | Gemma2-2B-it | Gemma2-9B-it | top-80% | 65.93 |
| ARC | Gemma2-2B-it | Gemma2-9B-it | top-70% | 57.12 |
| MGSM | Gemma2-2B-it | Gemma2-9B-it | top-90% | 62.97 |
| MGSM | Gemma2-2B-it | Gemma2-9B-it | top-80% | 70.43 |
| MGSM | Gemma2-2B-it | Gemma2-9B-it | top-70% | 82.25 |
| MMLU | Gemma2-2B-it | Gemma2-9B-it | top-90% | 68.39 |
| MMLU | Gemma2-2B-it | Gemma2-9B-it | top-80% | 74.25 |
| MMLU | Gemma2-2B-it | Gemma2-9B-it | top-70% | 82.88 |

Table 8: Mean Blue Scores between model pairs. All prompts across benchmarks are 5-shot and reused from the primary experiments described in Section 3.1.

## 5.3 How fast is LitePruner?

To understand the number of FLOPs needed by LitePruner, we ran LitePruner of randomly generated prompts of different lengths profiled for the number of floating-point operations(FLOPs) with Llama3-2b-it pruner and Gemma-2-2b-it pruners. Since, for any top-k% value, the core model does not change, there will be no change in the number of FLOPs. Also, due to several confounding factors such as other activities on the server, temperature of the hardware, different specifications of the hardware, etc., the time taken is an unreliable metric for speed in an experimental setup, and hence, it was not measured. However, it can be easily estimated on the basis of time complexity (Section

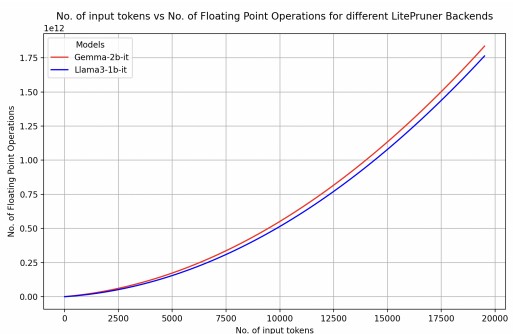

Figure 1: No. of FLOPs vs No. of input Tokens for Llama3-1b-t anfd Gemma-2-2b-it Lite Pruner Backends

2) and the number of FLOPs. The results can be seen in Figure 1. As observed, the graph follows the $O(n^2 d)$. It can also be seen that even for very long input lengths, LitePruner shows very few FLOPs for both backends. Llama3-1b-it backend has only 260 million parameters(less than 500m memory in bf16 setting), while Gemma-2-2b-it backend has only 590 million parameters (less than 1.2G memory in bf16), thus also demonstrating the low memory requirements for both models.

## 6 RELATED WORK

Token pruning focuses on reducing the number of tokens processed by models to save computational costs with minimal performance degradation. Currently, there are several methods that aim to address redundant token problems. Existing works, such as Ahia et al. (2023); Liang et al. (2023); Dewangan et al. (2025), focus on the development of entirely new tokenizers that allow fairer tokenization across languages. However, deploying these tokenizers remains challenging for pre-existing closed-weight models. Other methods, such as Huang et al. (2023), explore prompting techniques to improve performance without modifying the tokenizer. Several other works, including Xu et al. (2025), focus on improving the performance of LLMs in long-context scenarios through token pruning, although they do not specifically target multilingual settings with open-weight models. These methods typically prune tokens progressively layer-by-layer (Goyal et al., 2020b; Cao et al., 2023), which are different from our method. The recent prompt compressor family (Jiang et al., 2024; Pan et al., 2024) also pushes the efficiency idea further, but it is still requires running on large GPU memories. We consider the universal, real-life case that the input context could be pruned before passing it to the black-box APIs or LLMs without GPU support.

Our works share the same idea with (Goyal et al., 2020a; Ye et al., 2021) as we both leverage pre-trained attention weights for pruning. However, our off-the-shelf LitePruner is using a second model as the backend, making it distinguishable. Another parallel line is about token merging. Instead of removing some tokens, Bolya et al. (2022) suggest merging tokens to reduce the input length. Xing et al. (2024) attempt to merge similar or less informative tokens into summary representations. In our experiments, we found LitePruner can merge some short-length tokens into a longer token by removing and adjusting some neighboring tokens, showing some merging effects.

## 7 CONCLUSION

In this paper, we present LitePruner, a lightweight model to prune tokens before sending them to the target large models. We re-use minimal pre-trained weights from a small model to select important tokens but keep the relative token position unchanged. Massive in-context learning experiments on three multilingual benchmarks and RAG experiments show effectiveness of LitePruner. Meanwhile, LitePruner is compatible with commercial LLM APIs, contributing to practical applications. Long context, especially in multilingual settings, causes additional token fees, response latency, and long context processing. LitePruner attempts to improve the processing efficiency for long context by reducing the total number of tokens while maintaining decent performance in multilingual settings, especially for low-resource language.

ACKNOWLEDGMENTS

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

# A  APPENDIX

## A.1  PRELIMINARY

Our first question is whether LLMs can handle slightly broken inputs as our goal is to remove a small portion of tokens regardless of the input language surface and send the pruned input to the target large model. Recent studies have observed that LLMs can initiate internal de-tokenisation or reconstruction processes, wherein they deduce embedding representation of multi-token words Kaplan et al. (2025a); Kamoda et al. (2025). Kaplan et al. (2025a) further demonstrates that LLMs are capable of performing this reconstruction even for ill-formed words.

To study this question first, we conduct experiments for Llama3-1B-it and Llama3-3B-it on the multilingual MMLU dataset from Llama Evals Grattafiori et al. (2024) by performing a random token dropping process. Specifically, after tokenizing each input prompt, random tokens are dropped with a probability $p$. The last 10 tokens are excluded from this process to prevent inaccuracies during multiple-choice answer parsing, as only one token is allowed to be generated without any sampling.

In Table 9 and 10 , we observe a clear trend: dropping tokens leads to a significant reduction in the number of FLOPs, while causing only a relatively small degradation in accuracy for the these models. These results suggest that token pruning can serve as a promising method to decrease computational cost, while preserving comparable performance and improving efficiency. Meanwhile, when using commercial LLM APIs, which usually charge users by token usages, we can save budget if a query is pruned and has less tokens.

| Lang | $p$ | Avg. Input Tokens | FLOPs drop | Acc. Drop |
|------|-----|-------------------|------------|-----------|
| hi | 0.1 | 1614.4 | 10.0% | 2.8% |
|    | 0.2 | 1614.4 | 19.8% | 5.6% |
|    | 0.3 | 1257.9 | 26.9% | 7.0% |
|    | 0.4 | 1079.5 | 39.5% | 7.0% |
| th | 0.1 | 1404.9 | 10.8% | 3.5% |
|    | 0.2 | 1253.0 | 21.2% | 7.1% |
|    | 0.3 | 1100.4 | 31.7% | 7.8% |
|    | 0.4 | 950.0 | 41.7% | 9.8% |
| fr | 0.1 | 1404.9 | 10.8% | 3.5% |
|    | 0.2 | 1253.0 | 21.2% | 7.1% |
|    | 0.3 | 1100.4 | 31.7% | 7.8% |
|    | 0.4 | 950.0 | 41.7% | 9.8% |

Table 9: Effect of random token dropping for Llama3.2-1B-it on MMLU in 3 languages.

| Lang | $p$ | Avg. Input Tokens | FLOPs drop | Acc. drop |
|------|-----|-------------------|------------|-----------|
| hi | 0.1 | 1621.55 | 10.6% | 7.2% |
|    | 0.2 | 1442.39 | 18.5% | 11.9% |
|    | 0.3 | 1263.36 | 28.4% | 13.5% |
|    | 0.4 | 1085.27 | 38.2% | 15.4% |
| th | 0.1 | 1420.751 | 10.8% | 5.5% |
|    | 0.2 | 1267.125 | 21.3% | 13.6% |
|    | 0.3 | 1115.922 | 31.4% | 14.4% |
|    | 0.4 | 961.839 | 41.6% | 15.3% |
| fr | 0.1 | 1420.751 | 10.8% | 5.5% |
|    | 0.2 | 1267.125 | 21.3% | 13.6% |
|    | 0.3 | 1115.922 | 31.4% | 14.4% |
|    | 0.4 | 961.839 | 41.6% | 15.3% |

Table 10: Effect of random token dropping for Llama3.2-3B-it on MMLU in 3 languages.

| Layer No. | Multilingual ARC | | | MGSM | | | Global-MMLU-Lite | | |
|-----------|---|---|---|---|---|---|---|---|---|
|  | 1 | 2 | 3 | 1 | 2 | 3 | 1 | 2 | 3 |
| 1 | 0.00011235 | 0.0003288 | 0.0003304 | 0.0002371 | 0.0007176 | 0.000647 | 0.0002127 | 0.0006185 | 0.000638 |
| 2 | 0.0003765 | 4.67e-05 | 0.000942 | 6.574e-05 | 0.000141 | 6.574e-05 | 0.0.0007405 | 6.81e-05 | 0.000206 |
| 3 | 0.0004666 | 5.2e-05 | 5.025e-05 | 0.000986 | 0.00010383 | 0.0001616 | 0.000891 | 0.0001874 | 5.78e-05 |

Table 11: Average RAD values between layers of llama3-1B-it (y-axis) and llama3-3B-it (x-axis) for 1000 random prompts from each Global-MMLU-Lite and Multilingual ARC from Section 3.1.

| Layer No. | Multilingual ARC | | | MGSM | | | Global-MMLU-Lite | | |
|-----------|---|---|---|---|---|---|---|---|---|
|  | 1 | 2 | 3 | 1 | 2 | 3 | 1 | 2 | 3 |
| 1 | 0.9962 | 0.9596 | 0.9635 | 0.999 | 0.999 | 1.0 | 1.0 | 0.9995 | 1.0 |
| 2 | 0.9380 | 0.9982 | 0.9984 | 0.998 | 1.0 | 1.0 | 0.9995 | 0.9995 | 1.0 |
| 3 | 0.9424 | 0.9982 | 0.9986 | 0.998 | 1.0 | 1.0 | 0.9990 | 0.9995 | 1.0 |

Table 12: Average cosine similarities between layers of llama3-1B-it (y-axis) and llama3-3B-it (x-axis) for 1000 random prompts from each Global-MMLU-Lite, Multilingual ARC, and MGSM from Section 3.1.

| Layer No. | Multilingual ARC | | | MGSM | | | Global-MMLU-Lite | | |
|-----------|---|---|---|---|---|---|---|---|---|
|  | 1 | 2 | 3 | 1 | 2 | 3 | 1 | 2 | 3 |
| 1 | 0.9980 | 0.9990 | 0.9990 | 0.999 | 0.999 | 1.0 | 0.9995 | 1.00 | 1.0 |
| 2 | 0.9956 | 1.0 | 0.9995 | 0.998 | 1.000 | 1.0 | 0.9985 | 1.00 | 1.0 |
| 3 | 0.9956 | 1.0 | 1.0 | 0.998 | 1.000 | 1.0 | 0.9980 | 1.00 | 1.0 |

Table 13: Average cosine similarities between layers of llama3-1B-it (y-axis) and llama3-8B-it (x-axis) for 1000 random prompts from each Global-MMLU-Lite, Multilingual ARC, and MGSM from Section 3.1.

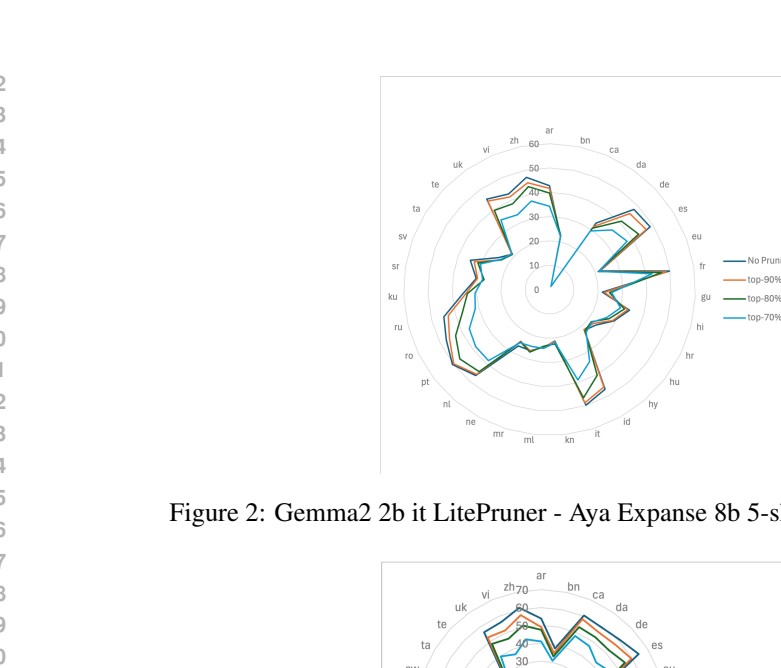

Figure 2: Gemma2 2b it LitePruner - Aya Expanse 8b 5-shot Multilingual ARC

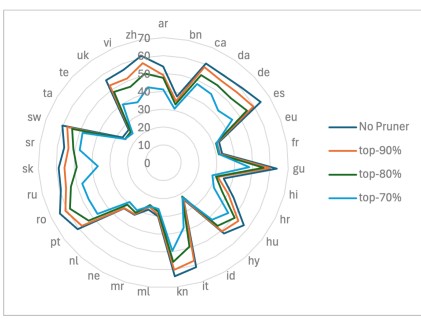

Figure 3: Gemma2 2b it LitePruner - Gemma 27b it Multilingual ARC

| Layer No. | Multilingual ARC | | | MGSM | | | Global-MMLU-Lite | | |
|---|---|---|---|---|---|---|---|---|---|
| | 1 | 2 | 3 | 1 | 2 | 3 | 1 | 2 | 3 |
| 1 | 0.9980 | 0.9980 | 0.9980 | 0.9990 | 0.9990 | 0.9985 | 0.9995 | 1.0 | 0.9995 |
| 2 | 0.9950 | 1.0 | 0.9995 | 0.9980 | 1.0 | 0.9995 | 0.9976 | 1.0 | 1.0 |
| 3 | 0.9960 | 1.0 | 1.0010 | 0.9980 | 1.0 | 1.0000 | 0.9976 | 1.0 | 1.0 |

Table 14: Average cosine similarities between layers of llama3-3B-it (y-axis) and llama3-3B-it (x-axis) for 1000 random prompts from each Global-MMLU-Lite, Multilingual ARC, and MGSM from Section 3.1.

| Layer No. | Multilingual ARC | | | MGSM | | | Global-MMLU-Lite | | |
|---|---|---|---|---|---|---|---|---|---|
| | 1 | 2 | 3 | 1 | 2 | 3 | 1 | 2 | 3 |
| 1 | 0.0001664 | 0.0003580 | 0.0004861 | 0.0003684 | 0.0008770 | 0.0010500 | 0.0003853 | 0.0007205 | 0.0009450 |
| 2 | 0.0005210 | 0.00006413 | 0.00006855 | 0.0011080 | 0.00006074 | 0.0001531 | 0.0010650 | 0.00008446 | 0.0001662 |
| 3 | 0.0005220 | 0.00006560 | 0.00006783 | 0.0010360 | 0.00010127 | 0.0002189 | 0.0010840 | 0.0001140 | 0.0001450 |

Table 15: Average RAD values between layers of llama3-3B-it (y-axis) and llama3-8B-it (x-axis) for 1000 random prompts from each Global-MMLU-Lite, Multilingual ARC, and MGSM from Section 3.1.

## A.2 PERFORMANCE FOR LANGUAGES

## A.3 LAYER COMPARISON

## A.4 CASE STUDY

We show an example of LitePruner's result in Figure 14.

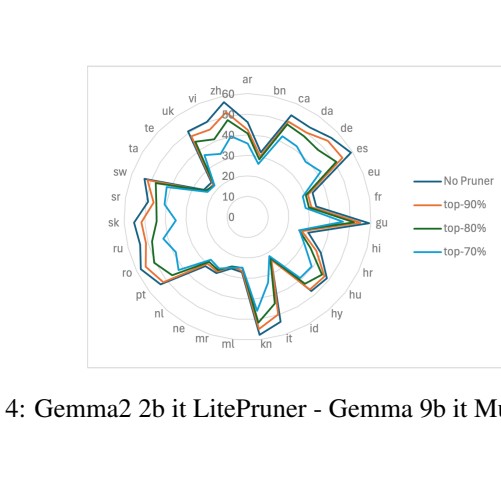

Figure 4: Gemma2 2b it LitePruner - Gemma 9b it Multilingual ARC

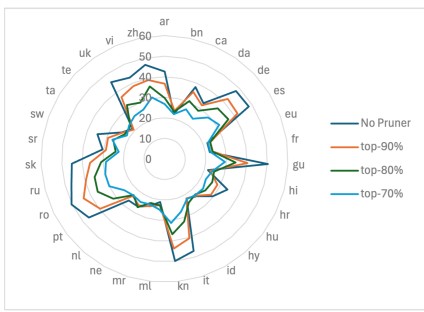

Figure 5: Llama 1b it LitePruner - Aya Expanse 8b 5-shot Multilingual ARC

| Layer No. | Multilingual ARC | | | MGSM | | | Global-MMLU-Lite | | |
|---|---|---|---|---|---|---|---|---|---|
| | 1 | 2 | 3 | 1 | 2 | 3 | 1 | 2 | 3 |
| 1 | 0.0001957 | 0.0002687 | 0.0003958 | 0.0004778 | 0.0006860 | 0.0008574 | 0.0004702 | 0.0005530 | 0.0007760 |
| 2 | 0.0004787 | 2.55E-05 | 0.0001103 | 0.0011410 | 7.86E-05 | 0.0001240 | 0.0010195 | 4.47E-05 | 0.0002086 |
| 3 | 0.0005690 | 0.00011015 | 2.17E-05 | 0.0011835 | 0.0001286 | 7.38E-05 | 0.0011720 | 0.0001892 | 5.95E-05 |

Table 16: Average RAD values between layers of llama3-1B-it (y-axis) and llama3-8B-it (x-axis) for 1000 random prompts from each Global-MMLU-Lite, Multilingual ARC, and MGSM from Section 3.1.

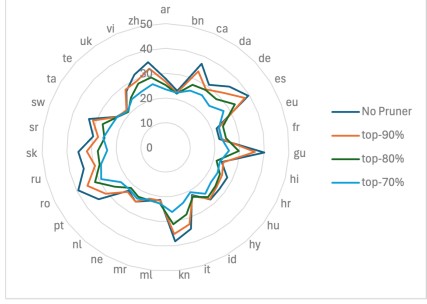

Figure 6: Llama 1b it LitePruner - Llama 3b it 5-shot Multilingual ARC

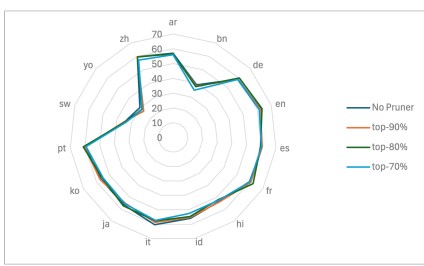

Figure 7: Llama 1b it LitePruner - Llama 8b it 5-shot Multilingual ARC

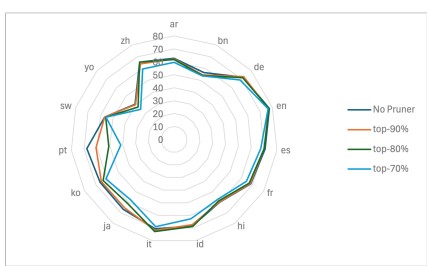

Figure 8: Llama 1b it LitePruner - Aya Expanse 8b 5-shot Global-MMLU-Lite

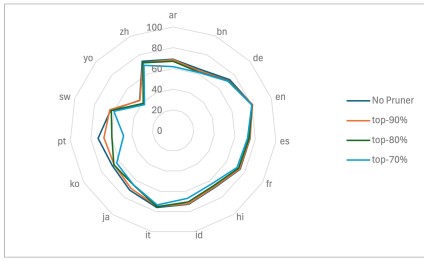

Figure 9: Gemma2 2b it LitePruner - Gemma2 9b it 5-shot Global-MMLU-Lite

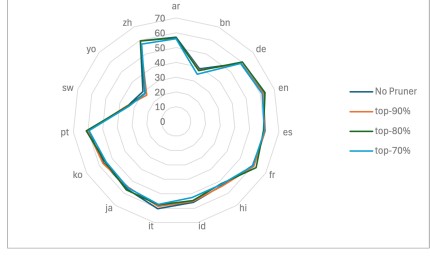

Figure 10: Gemma2 2b it LitePruner - Gemma2 27b it 5-shot Global-MMLU-Lite

Figure 11: Gemma2 2b it LitePruner - Aya Expanse 8b 5-shot Global-MMLU-Lite

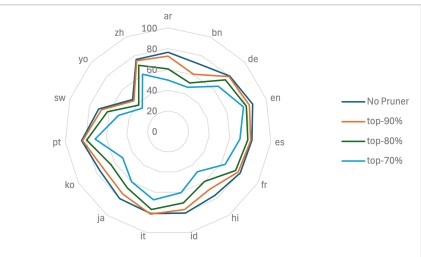

Figure 12: Llama 1b it LitePruner - Llama 70b it 5-shot Global-MMLU-Lite

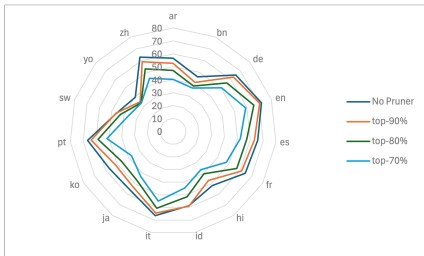

Figure 13: Llama 1b it LitePruner - Llama 8b it 5-shot Global-MMLU-Lite

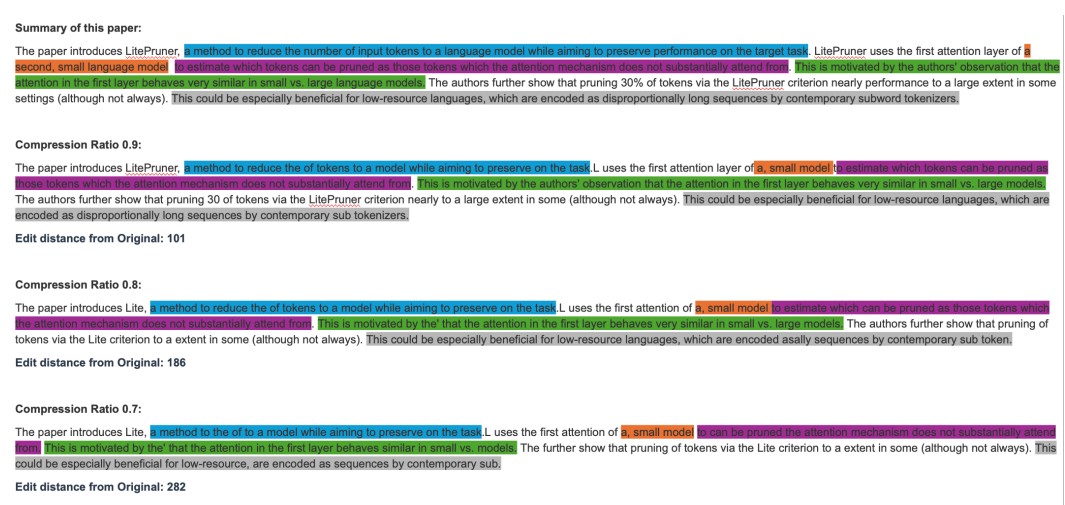

Figure 14: Case Study

