# OpenReview forum: "LitePruner: A Lightweight Realtime Token Pruner before Large Language Models"
_ICLR.cc/2026/Conference — ICLR 2026 Conference Withdrawn Submission_

### Official Review · Reviewer_KCka · 2025-10-19

**Soundness:** 2
**Presentation:** 2
**Contribution:** 2
**Rating:** 2
**Confidence:** 3

**Summary:**

This paper proposes using the first attention layer of small pre-trained models to rank and prune input tokens before passing them to larger LLMs, motivated by tokenization disparities in multilingual settings. The method is training-free, CPU-based, and evaluated on multilingual ICL and RAG benchmarks with reported token reductions of up to 30%.

**Strengths:**

1. The paper addresses a real fairness issue where non-English users pay significantly more for LLM services due to tokenization disparities, and provides a practical deployment that is CPU-based, training-free, and compatible with commercial APIs.

2. The paper provides extensive experiments across multiple benchmarks, languages (high/medium/low-resource), and model families.

**Weaknesses:**

1. The paper claims that "Early layers in both small and large models show similar attention patterns due to similar tokenization", and provided evidence (Tables 6-7) of high cosine similarity between attention distributions.However, correlation does nto equate to causation. If would be great if the authors can show that tokens with low attention in the small model are actually redundant for the large model. High cosine similarity just means the OOD attention patterns correlate, and doesn't validate that these patterns predict what can be pruned.

2. Llama3-70B improves from 4.0% to 31.2%. This isn't noise removal, as the performance even after the improvement is lackluster.

**Questions:**

1. What happens with causal mask?

2. How do you compare to frequency-based baseline?

3. What explains the MGSM anomalies?

---

> ### Author Response · Authors · 2025-12-03
>
> **W1: The paper claims that "Early layers in both small and large models show similar attention patterns due to similar tokenization", and provided evidence (Tables 6-7) of high cosine similarity between attention distributions.However, correlation does not equate to causation. It would be great if the authors can show that tokens with low attention in the small model are actually redundant for the large model. High cosine similarity just means the OOD attention patterns correlate, and doesn't validate that these patterns predict what can be pruned.**
>
> We have shown analysis with the BLEU scores showing very high similarity between the large model and the small model. This indicates that the large model and the small model yield similar attention patterns in the first layer, justifying our design and choice.
>
> **W2: Llama3-70B improves from 4.0% to 31.2%. This isn't noise removal, as the performance even after the improvement is lackluster.**
>
> [CORRECTION] It is 3.2% not 31.2. There is no improvment for this experiment.
>
> **Questions:**
>
> - What happens with a causal mask?
>
> No causal mask is used while pruning. Causal masks are used in the target models as we send texts to them.
>
> - How do you compare to frequency-based baseline?
>
> Frequency-based baselines require additional resouces and memories, and it is not suitable to low-resouce languages.
>
> - What explains the MGSM anomalies?
>
> Unfortunately, we have a typo here. This might refer to W2.

---

### Official Review · Reviewer_mNvA · 2025-10-28

**Soundness:** 3
**Presentation:** 2
**Contribution:** 3
**Rating:** 4
**Confidence:** 2

**Summary:**

This paper proposes LitePruner, a training-free, CPU-based token pruning method that reduces input token counts in multilingual settings while preserving downstream performance. It leverages relative attention weights (RAW) from the first attention layer of a small pretrained model to rank token importance and forwards only the top-k% tokens to the target large model. The authors validate effectiveness on multilingual ICL (MGSM, Global-MMLU-Lite, ARC) and RAG (PubMedQA, MEMERAG) benchmarks, demonstrating generalization across both in-family and across-family model settings.

**Strengths:**

LitePruner is practical. It requires no training, runs on CPU, and can be deployed before commercial APIs like GPT to save token costs. Experiments span multiple languages, model families (Llama3, Gemma2, Aya), and benchmarks (MGSM, ARC, MMLU, PubMedQA). Results show minimal performance drop at top-90% pruning, and even improvements in some low-resource languages, suggesting potential denoising effects. The authors also provide empirical support via RAD and cosine similarity, showing early-layer attention alignment between small and large models.

**Weaknesses:**

The paper provides insufficient theoretical justification for its core assumption that the first attention layer alone captures sufficient token importance, particularly for complex multilingual reasoning tasks such as MGSM with chain-of-thought prompting. While the authors present empirical correlations using RAD and cosine similarity, they do not explain why early-layer attention should generalize across tasks, languages, or model families. Methodologically, Algorithm 1 omits key implementation details; for example, it does not specify how positional encodings are adjusted after arbitrary token removal, which may affect the target model’s interpretation of sequence order. The method’s reliance on raw attention weights from the first layer also makes it incompatible with efficient operators such as Flash Attention, limiting its practical deployment efficiency despite the claimed CPU compatibility. In the RAG experiments, only documents are pruned while queries remain intact, but the paper offers no rationale for this asymmetric treatment, which could influence retrieval quality.

**Questions:**

N/A

---

> ### Author Response · Authors · 2025-12-03
>
> **W1: The paper provides insufficient theoretical justification for its core assumption that the first attention layer alone captures sufficient token importance, particularly for complex multilingual reasoning tasks such as MGSM with chain-of-thought prompting.**
>
> LLMs utilize similar frameworks to train tokenizers (e.g., BPE). These frameworks merge frequent tokens/subwords in the training corpus. This is the key insight for the across-family setting. While different LLMs may use different corpora, and tokenizer may utilize additional rules (sentence-piece vs word-piece), frequent tokens/subwords are similar because of Zifp’s law.
>
> In line 53, we also state "In contrast, we hypothesize that for the same context, in the early layers, both small and
> large models potentially show similar attention patterns because the early layers latch onto the same shallow local signals and attempt to restore words from subtokens (i.e., detokenization (Kaplan et al.,2025b)) due to the similar tokenization algorithm and the same attention mechanism."
>
> **W2: While the authors present empirical correlations using RAD and cosine similarity, they do not explain why early-layer attention should generalize across tasks, languages, or model families.**
>
> We would like to refer to the reponse for W1.
>
> **W3: Methodologically, Algorithm 1 omits key implementation details; for example, it does not specify how positional encodings are adjusted after arbitrary token removal, which may affect the target model’s interpretation of sequence order.**
>
> The tokens removed by LitePruner are based on the attention weights and are not arbitrarily pruned out. Then the remaining tokens in their relative original order are tokenized and retokenized and fed to the main LLM with causal mask and new positional encoding as per new retokenization.
>
> **W4: The method’s reliance on raw attention weights from the first layer also makes it incompatible with efficient operators such as Flash Attention, limiting its practical deployment efficiency despite the claimed CPU compatibility.**
>
> We argue that this is not a limitation of our method. The weights, including linear projections in attention, embeddings, position encodings, and embedding normalization, can be allocated on either the CPU (preferred) or GPU, and we can compute attention scores normally, following the mathematical formula for attention.
>
> **W5: In the RAG experiments, only documents are pruned while queries remain intact, but the paper offers no rationale for this asymmetric treatment, which could influence retrieval quality.**
>
> One reason is that the search query is very small and is usually dense in information and can lead to loss of information when token pruning while documents have many more tokens that could potentially allow LitePruner to select tokens.

---

### Official Review · Reviewer_VbKi · 2025-10-28

**Soundness:** 3
**Presentation:** 3
**Contribution:** 2
**Rating:** 2
**Confidence:** 4

**Summary:**

The paper presents a training free framework to rank token importance, which prunes the context of the input and sends only a subset of important tokens to the target model using a computationally cheap model. It shows that this method can achieve competitive results for low-resource language tasks and save token budgets due to less efficient tokenization for these texts.

**Strengths:**

1. The author targets an important question and proposes a conceptually simple yet effective framework for solving the problem.
2. It has been tested on several benchmarks, making it very comprehensive.
3. It leverages the clear motivation that the small and large models share similar attention patterns, which can be used as a proxy for token importance estimation.

**Weaknesses:**

1. Efficiently solving low-resource language tasks is a very interesting problem, which should be the focus of the paper as suggested in the abstract and intro. However, very little room has been left for discussing the specific characteristics of these tasks (e.g. tokenization fairness as mentioned by the author). A very straightforward baseline that can be potentially included is to see if a direct translation of the prompt from low to high resource language can improve efficiency and quality.
2. There are many similar prior works which use the same method and has been well tested on many long context benchmarks (e.g. SpecPrefill https://arxiv.org/abs/2502.02789 being the most similar one). This should be cited and compared accordingly since many aspects of the paper shares the same insights and methodology as SpecPrefill (e.g. use a smaller draft model to prune important tokens, use attention as the surrogate, the handling of position ids, etc). Another similar line of work is called GemFilter, which uses the model’s own shallow layers as proxy.
3. How does this method work in the multi-turn setting? This should be a potential pitfall for this method. If not, it should be discussed clearly.
4. On line 258, the author mentions that “top-90% is still a common choice for all scenarios”. The reviewer thinks that keeping 90% would be a relatively high value for this method to break even the cost of pruning itself. This should be discussed more formally. Since the method should either 1) improve accuracy or 2) increase the efficiency.

**Questions:**

Listed above in weakness.

---

> ### Author Response · Authors · 2025-12-03
>
> **W1: Efficiently solving low-resource language tasks is a very interesting problem, which should be the focus of the paper as suggested in the abstract and intro. However, very little room has been left for discussing the specific characteristics of these tasks (e.g. tokenization fairness as mentioned by the author). A very straightforward baseline that can be potentially included is to see if a direct translation of the prompt from low to high resource language can improve efficiency and quality.**
>
> Translation of prompts would require more computation to translate them to another language than pruning tokens as translation would require running a sizable model for accurate translation. Our method only needs only the first attention layer of a small model which is very small compared to a translation model. At the same time our method shows improvement in English as well in RAG scenarios(PUbMedQA), which makes it more generalizable than translation.
>
>
> **W2:There are many similar prior works which use the same method and has been well tested on many long context benchmarks (e.g. SpecPrefill https://arxiv.org/abs/2502.02789 being the most similar one). This should be cited and compared accordingly since many aspects of the paper shares the same insights and methodology as SpecPrefill (e.g. use a smaller draft model to prune important tokens, use attention as the surrogate, the handling of position ids, etc). Another similar line of work is called GemFilter, which uses the model’s own shallow layers as proxy.**
>
> While it is true that several  methods like GemFilter and Learned Token Pruning  own attention layers to filter tokens within the models, for commercial APIs and Blackbox models, it is not possible to carry out the process as the inner model weights and architecture cannot be accessed.  LitePruner solves this issue as it only needs small open source models. In orther words, we have different motivations and applications.
>
> **W3:How does this method work in the multi-turn setting? This should be a potential pitfall for this method. If not, it should be discussed clearly.**
>
> Even in Multi-turn settings, since the input to the main LLM is only opened once, it still allows K-V caching which does not lead to any issues.
>
> **W4: On line 258, the author mentions that “top-90% is still a common choice for all scenarios”. The reviewer thinks that keeping 90% would be a relatively high value for this method to break even the cost of pruning itself. This should be discussed more formally. Since the method should either 1) improve accuracy or 2) increase the efficiency.**
>
> Thanks for pointing out this. Our goal is to reduce the api fees (by reducing tokens) and attempt to preserve the performance as stated in abstract and introduction (e.g., line 63).

---

### Official Review · Reviewer_rRRh · 2025-10-29

**Soundness:** 2
**Presentation:** 2
**Contribution:** 2
**Rating:** 4
**Confidence:** 3

**Summary:**

The paper addresses the problem of tokenization inefficiency and fairness in multilingual large language models, where non-English languages produce disproportionately more tokens, leading to higher costs, slower inference, and shorter usable context. It proposes LitePruner, a training-free, CPU-based token pruning method that reuses the embedding and first attention layer of a small pre-trained model to rank token importance and remove less important tokens before sending input to the target large model.

The authors conduct two main sets of experiments. First, in-context learning (ICL) tests are performed on multilingual benchmarks MGSM, Global-MMLU-Lite, and Multilingual ARC, under both in-family (e.g., Llama3-1B for Llama3-70B, Gemma2-2B for Gemma2-27B) and across-family (e.g., Llama3-1B for GPT-4.1-nano, Gemma2-2B for Aya-expanse-8B) settings, using 3-, 5-, and 8-shot prompting. Second, retrieval-augmented generation (RAG) experiments are conducted on PubMedQA and MEMERAG, where documents are pruned by LitePruner before retrieval and evaluated with metrics such as Mean Reciprocal Rank (MRR), Faithfulness (FA), and Semantic Answer Similarity (SAS).

**Strengths:**

- The paper’s idea of performing pre-inference token pruning using the first attention layer of a small model is moderately original. Although adopt small model as a proxy of the attention score is not a novel idea, applying this method in permanent token pruning in LLMs is new.
- Experiments are conducted under multiple settings, covering both in-family and across-family settings across multiple model families (Llama 1/8B, Gemma, GPT-4.1-nano) and two task types (in-context learning and retrieval-augmented generation).

**Weaknesses:**

- The method relies on shared tokenization between the small and large models (e.g., Llama3-1B for Llama3-70B). However, in cross-family settings (e.g., Gemma for GPT-4.1-nano), tokenizers and embedding spaces differ substantially, which may undermine the assumption of attention pattern similarity.
- Some detailed hyperparameter choices (e.g., head averaging details, normalization methods, or handling of positional encodings after pruning) are not specified.
- The evaluation results are not consistently stable across datasets, and the underlying reasons are not sufficiently analyzed. For instance, the performance drops observed on MGSM and Global-MMLU-Lite are notably large and remain unexplained.
- In the in-family and cross-family experiments, different large language models are used as targets, making it difficult to directly assess how the quality of the small model influences the pruning results.
- No explicit ablation study is conducted to test the sensitivity or necessity of specific design choices in LitePruner.

**Questions:**

Please refer to the weaknesses part.

---

> ### Author Response · Authors · 2025-12-03
>
> **W1: The method relies on shared tokenization between the small and large models (e.g., Llama3-1B for Llama3-70B). However, in cross-family settings (e.g., Gemma for GPT-4.1-nano), tokenizers and embedding spaces differ substantially, which may undermine the assumption of attention pattern similarity.**
>
> LLMs utilize similar frameworks to train tokenizers (e.g., BPE). These frameworks merge  frequent tokens/subwords in the training corpus. This is the key insight for the across-family setting. While different LLMs may use different corpora, and tokenizer may utilize additional rules (sentence-piece vs word-piece), frequent tokens/subwords are similar because of Zifp’s law. For example, we can tokenize the abstract of this paper via LLama’s tokenizer and Gemma’s tokenizer. The results are below:
>
>
> Llama:
> 'Token ization is one of the core steps of the language model pipeline . However , the tokenizer yields more tokens for the same context in non - English languages , especially in low -resource languages due to the shared mult ilingual settings , which results in unexpected fairness problems in terms of token fees , response latency , and long context processing . In this paper , we study a real -time computing problem , attempting to reduce the total number of tokens per query but maintain decent performance in mult ilingual settings . We present a simple , training -free , CPU -based pr uner model to reuse pre -trained weights from the first attention layer of small models to rank token importance , only delivering important tokens to the target larger models . This method is motivated by the fact that early layers in both small and large models latch onto similar shallow local signals due to similar token ization algorithms ( e .g ., B PE ) producing identical local signals . Massive in -context learning experiments on MG SM , Global -M ML U -L ite and ARC and R AG -based experiments on PubMed QA and MEM ER AG show that our method can preserve decent performance for languages while reducing up to $ 30 \\ % $ of the total number of tokens in both in -family and across -family model settings , where the pr uner model and the target large model are in or not in the same model family . Our method is compatible with commercial L LM APIs and CPU -based , contributing to real -life applications . '
>
> Gemma:
> 'Token ization is one of the core steps of the language model pipeline . However , the tokenizer yields more tokens for the same context in non - English languages , especially in low - resource languages due to the shared multilingual settings , which results in unexpected fairness problems in terms of token fees , response latency , and long context processing . In this paper , we study a real - time computing problem , attempting to reduce the total number of tokens per query but maintain decent performance in multilingual settings . We present a simple , training - free , CPU - based pr uner model to reuse pre - trained weights from the first attention layer of small models to rank token importance , only delivering important tokens to the target larger models . This method is motivated by the fact that early layers in both small and large models latch onto similar shallow local signals due to similar token ization algorithms ( e . g ., B PE ) producing identical local signals . Massive in - context learning experiments on MG SM , Global - M ML U - Lite and ARC and RAG - based experiments on PubMed QA and MEM ER AG show that our method can preserve decent performance for languages while reducing up to $ 3 0 \\% $ of the total number of tokens in both in - family and across - family model settings , where the pr uner model and the target large model are in or not in the same model family . Our method is compatible with commercial L LM APIs and CPU - based , contributing to real - life applications . '
>
> The edit distance is only 20, and thus there is no substantiate difference.

---

> ### Author Response · Authors · 2025-12-03
>
> **W2: Some detailed hyperparameter choices (e.g., head averaging details, normalization methods, or handling of positional encodings after pruning) are not specified.**
>
> - head averaging details:  In line 104, we state “Let [h, n , n] to denote the multi-head attention score matrix with h heads for the input $X_n$ with n tokens. Then, we define the importance score for i-th token $x_i \in X_n$ as $IS(x_i) = avg([h, n , i])$”, so we average for all heads.
>
> - normalization methods: we do not utilize an additional normalization. The softmax used in attention could offer “normalized” scores.
>
> - handling of positional encodings after pruning:  In line 129, we state “ Note that we do not change the relative positions for all tokens. However, the absolute positions are modified as some tokens are removed. After pruning, the target LLM can add position encoding normally as we deliver the text string to it.” For example (the same to Figure 15 in the last page of this paper), when inputing x=“This paper introduces LitePruner, a method to reduce the number of input tokens to a language model while aiming to preserve performance on the target task.”, our LitePruner produce x_pruned=“The paper introduces Lite, a method to reduce the of tokens to a model while aiming to preserve on the task.”, and we input x’ to the target model.  In implementation, this can be achieved by decode/detokenize lookup ids back to the text string (Algorithm 1 from  line 16-21). Therefore, we don’t have to handle positional encodings after pruning, and the target model will process x’ with normal positional encodings.
>
> **W3: The evaluation results are not consistently stable across datasets, and the underlying reasons are not sufficiently analyzed. For instance, the performance drops observed on MGSM and Global-MMLU-Lite are notably large and remain unexplained.**
>
> We would like to clarify this. For each task/dataset, the results are consistent across different settings and models.  For the  cross-dataset comparison,  we do observe large drops and argue that comparison is not fair in these settings because different tasks/datasets have different features.
>
> **W4:In the in-family and cross-family experiments, different large language models are used as targets, making it difficult to directly assess how the quality of the small model influences the pruning results.**
>
> We use the different target models because the across-family experiments are used to show LitePruner is not dependent on the same model  family. To show how the quality of the small model influences the pruning results, in Table 3, we use two different small models to prune the same text for the same target model.
>
> **W5:No explicit ablation study is conducted to test the sensitivity or necessity of specific design choices in LitePruner.**
>
> The key factor in our design is the choice of layers. To do the ablation study, we vary the layers and analyze the results in Table 6 and 7.

---

### Note · Authors · 2026-01-05

**Comment:**

Thanks for providing valuble and actionable suggestions.

**Withdrawal Confirmation:**

I have read and agree with the venue's withdrawal policy on behalf of myself and my co-authors.